# Consistent Story Generation: Unlocking the Potential of Zigzag Sampling

**Mingxiao Li**
KU Leuven
mingxiao.li@kuleuven.be

**Mang Ning**
Utrecht University
m.ning@uu.nl

**Marie-Francine Moens**
KU Leuven
sien.moens@kuleuven.be

## Abstract

Text-to-image generation models have made significant progress in producing high-quality images from textual descriptions, yet they continue to struggle with maintaining subject consistency across multiple images, a fundamental requirement for visual storytelling. Existing methods attempt to address this by either fine-tuning models on large-scale story visualization datasets, which is resource-intensive, or by using training-free techniques that share information across generations, which still yield limited success. In this paper, we introduce a novel training-free sampling strategy called Zigzag Sampling with Asymmetric Prompts and Visual Sharing to enhance subject consistency in visual story generation. Our approach proposes a zigzag sampling mechanism that alternates between asymmetric prompting to retain subject characteristics, while a visual sharing module transfers visual cues across generated images to enforce consistency. Experimental results, based on both quantitative metrics and qualitative evaluations, demonstrate that our method significantly outperforms previous approaches in generating coherent and consistent visual stories. The code is available at `https://github.com/Mingxiao-Li/Asymmetry-Zigzag-StoryDiffusion`.

## 1 Introduction

In recent years, breakthroughs in visual generation techniques have fundamentally transformed the way visual content is created. State-of-the-art image [1–4] and video generation models [5–10] now enable users to produce highly diverse visual outputs with remarkable realism and flexibility. These models can be guided by various forms of control, including bounding boxes [11, 12], object motion trajectories [13, 14], image prompts [5, 4], and even brain signal inputs [15–17], significantly expanding the creative possibilities in both professional and everyday settings.

Despite these impressive advances, challenges remain, particularly in visual storytelling tasks, where multiple pieces of visual content must consistently preserve the identity and key characteristics of subjects across scenes, three examples are presented in Figure 1. A dominant approach to this problem is personalization, which involves learning a subject-specific embedding [18–23]. While effective, this method has notable limitations: the need for per-subject fine-tuning makes it resource-intensive and difficult to scale, and it often leads to overfitting, which can degrade the model's general understanding of text and reduce its generative diversity. To address these limitations, past research has explored tuning-free methods [24–26], which encode subject information using a unified image encoder and leverage large-scale pretraining. These methods eliminate the need for per-subject optimization and offer greater scalability. However, they rely on high-quality, large-scale personalization datasets that are costly to curate and demand substantial computational resources for effective training.

More recently, a line of training-free methods has emerged, targeting the specific challenge of subject-consistent visual story generation without fine-tuning or extensive subject-specific data. The

39th Conference on Neural Information Processing Systems (NeurIPS 2025).

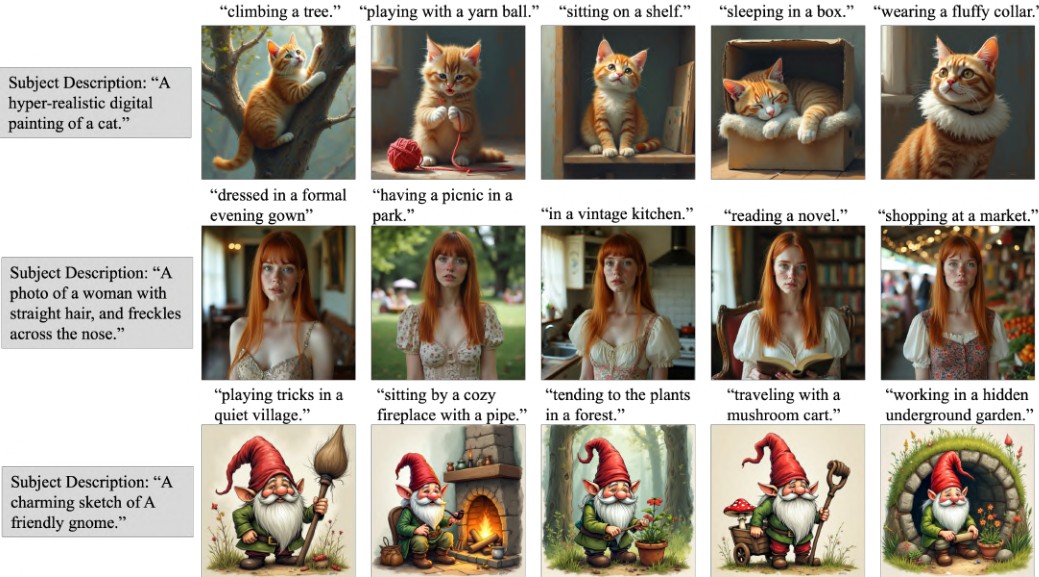

Figure 1: **Visual Story Telling** requires maintaining subject consistency across a sequence of generated images while ensuring that each image faithfully reflects the corresponding prompt. (Images generated using the FLUX model with our proposed method.)

StoryDiffusion model [27] introduces consistent attention by constructing image-level interactions through the random sharing of visual features across different frames. In contrast, the ConsistentStory model [28] aims to preserve subject identity by establishing visual interactions between images more selectively. However, while both methods show promise in maintaining subject consistency, they suffer from notable drawbacks: a reduced ability to follow textual prompts and diminished generative diversity. These issues arise from their direct interference with the diffusion model's generative process. Departing from these strategies, the One-Prompt One-Story model [29] proposes a different approach. First, it fuses all the prompts from a story into a single extended prompt. Then, it generates each image by reweighting the prompt embeddings to reflect the desired focus for each scene. In addition, it improves text-to-image attention by reinforcing subject-relevant information within the prompt itself. Although this model achieves good prompt fidelity, it struggles to maintain subject consistency, particularly when the prompt offers limited detail about the subject.

The above approaches show that directly sharing visual features during the generative process tends to weaken the model's ability to follow textual prompts, while relying solely on prompts to maintain subject consistency proves insufficient. To address both issues, we propose a novel method: **A**symmetry **Z**igzag **S**ampling (AZS). Unlike prior methods that inject subject-specific information into the intermediate representations of each generated image, our approach focuses on incorporating subject information directly into the latent representations of each scene. This distinction allows for more effective control over subject consistency while preserving alignment with textual prompts. As illustrated in Figure 2, our method strategically leverages latent-level visual sharing to improve narrative coherence in multi-scene generation.

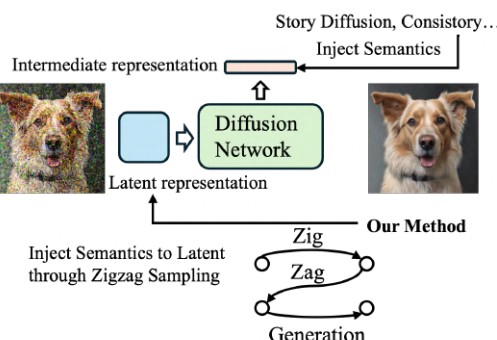

Figure 2: An illustration of Zigzag Sampling, along with a comparison to previous methods in how semantic information is incorporated during image generation.

As shown in Figure 2, our method decomposes each generation step into three sub-steps—zig, zag, and generation. It proposes the combination

of two key components: Zig Visual Sharing (ZVS) and Asymmetric Prompt Zigzag Inference (APZI). In the ZVS module, subject-specific visual information that is extracted using a subject-only prompt is injected into the self-attention layers during the generation of each scene. This allows the model to encode subject identity directly into the latent representations without disrupting overall scene composition. To further balance subject consistency with textual fidelity, APZI introduces an asymmetric prompt schedule: the zig and generation steps leverage the one-prompt technique proposed in [29], while the zag step employs a null prompt. This asymmetry prevents overfitting to textual descriptions and provides the model with a dedicated phase to integrate subject information. Together, ZVS and APZI enable effective subject conditioning in the latent space while preserving the model's ability to follow textual prompts, resulting in more coherent and consistent multiscene image generation.

We evaluate our method on two representative text-to-image architectures: the U-Net-based SDXL model [30] and the transformer-based FLUX model [31]. Experimental results demonstrate that our approach significantly improves subject consistency in generated images while maintaining strong alignment with textual prompts, outperforming existing baselines. In summary, our main contributions are:

- We propose a novel asymmetric zigzag sampling algorithm that injects visual semantics during the zig step to enforce subject consistency, uses a null prompt in the zag step to refine the latent space, and applies only text guidance in the generation step to preserve narrative coherence.
- We conduct extensive experiments and benchmark our approach against a range of existing methods to demonstrate its superiority.
- We validate the generalizability of our method by applying it to two widely used text-to-image architectures.

## 2 Related Work

### 2.1 Diffusion Models

Diffusion models are a class of generative models that have demonstrated remarkable success in synthesizing high-quality images [1, 30, 4, 3], videos [9, 7, 5, 6], and 3D content [32–34]. These models consist of a forward (noising) process and a reverse (denoising) process. In the forward process, a clean image is progressively corrupted by the addition of Gaussian noise over several steps. During the reverse process, a neural network learns to gradually denoise a random Gaussian sample, ultimately reconstructing a coherent image. The multi-step nature of this inference pipeline offers significant flexibility for controllable or guided generation tasks, such as layout-constrained image synthesis [12, 35, 36, 11], motion-controlled video generation [13, 14], and subject-consistent image personalization [19, 18]. Despite their strong performance, diffusion models remain an active area of research, with ongoing efforts to expand their generative capabilities and improve efficiency. For instance, recent works [37, 38] have proposed accelerated sampling strategies that reduce the number of required inference steps to fewer than 50. Other studies address issues such as exposure bias [39, 40], leading to more robust and higher-fidelity outputs. In general, diffusion models continue to evolve rapidly, and innovations in architecture, training strategies, and sampling techniques push the boundaries of generative modeling.

### 2.2 Training Based Consistent Text to Image Generation

Since the release of powerful open-source text-to-image diffusion models, generating images featuring consistent subjects — whether for image personalization or visual storytelling — has become an increasingly active area of research. Early approaches [18, 19, 41, 23, 20–22] primarily rely on test-time tuning techniques. These methods typically fine-tune either the entire diffusion network or specific components, learning a unique subject embedding or a small auxiliary network from a few reference images. Although these methods achieve high-quality results in subject-driven image generation, their reliance on test-time optimization poses significant scalability challenges. To address this limitation, a growing body of work [42, 25, 43–47] has explored encoder-based strategies. These methods utilize image encoders to extract subject-specific features from reference images and inject

the resulting embeddings into the diffusion model to guide subject-consistent generation. By avoiding per-subject fine-tuning, such approaches improve efficiency and scalability in real-world applications.

## 2.3 Training-Free Consistent Text to Image Generation

In parallel with training-based approaches, training-free methods for consistent text-to-image generation have also received significant attention in the visual generation community. A popular direction involves leveraging the attention mechanism to identify subject-relevant visual features, which are then reused to guide subsequent image generation. Initially, such techniques were explored in tasks like appearance transfer [48], image editing [49, 50], and image translation [51]. More recently, ConsisStory [28] adapted this idea to visual storytelling. Specifically, it uses cross-attention scores between text and image tokens to identify subject-relevant visual features in one image, and then injects these features into the generation of subsequent images to maintain subject consistency. In contrast to attention-based guidance, the current state-of-the-art method, 1Prompt1Story [29], maintains subject consistency by composing a unified prompt that combines all scene descriptions. During generation, the model adjusts the weighting of each sub-prompt depending on the scene. Our approach differs from both strategies by introducing **asymmetric guidance** within the zigzag sampling framework. This asymmetric design strengthens subject consistency across generated images while preserving the model's ability to follow textual prompts faithfully.

# 3 Method

## 3.1 Preliminary

**Latent Diffusion Model.** The Latent Diffusion Model (LDM) comprises an autoencoder—consisting of an encoder $\mathcal{E}$ and a decoder $\mathcal{D}$—that maps between pixel space and latent space. A diffusion model $\epsilon_\theta$, parameterized by $\theta$, is trained to model the noise in the latent space. For text-conditioned generation, a frozen text encoder $\tau_\zeta$ is used to embed the input text $\mathcal{P}$ into a dense representation [4]. The diffusion model is trained using the following loss function:

$$L_{LDM} = \mathbb{E}_{c \sim \epsilon(x), \epsilon \sim \mathcal{N}(0,1), t \sim \text{Uniform}(1,T)} \big[ ||\epsilon - \epsilon_\theta(z_t, t, \tau_\zeta(\mathcal{P}))||_2^2 \big] \tag{1}$$

During training, the diffusion process involves predicting noise at a randomly sampled time step $t$, drawn from a uniform distribution. To effectively model this denoising process—especially in the context of text-to-image generation—attention mechanisms play a central role in the architecture of diffusion models. In UNet-based models such as SDXL [30], cross-attention layers facilitate the integration of textual information by aligning latent visual features with text embeddings. This allows the model to generate images that are semantically consistent with the input text. Simultaneously, self-attention layers help capture spatial and semantic relationships within the visual latent space, enabling more coherent and detailed image synthesis. In contrast, the transformer-based FLUX model [31] adopts a different strategy. It uses a unified self-attention mechanism with modality-specific projection layers for visual and textual inputs. This design allows FLUX to fuse semantic information across modalities without relying on explicit cross-attention, leveraging the transformer's strength in modeling complex dependencies. Our method is seamlessly integrated in the UNet-based model SDXL and the transformer-based model FLUX.

## 3.2 Asymmetry Zigzag Sampling.

To address the challenge of balancing subject consistency and text fidelity in story-based image generation, we propose a novel **Asymmetric Zigzag Sampling** strategy that leverages the strengths of both diffusion-based generation and semantic conditioning. Prior work has introduced zigzag sampling—a method that decomposes each diffusion denoising step into three sub-steps: zig, zag, and generation—to improve the performance of generative models. Inspired by the distinct functional roles identified in recent work [52], where the zig step facilitates exploration, the zag step enables refinement, and the generation step produces the final output, we introduce asymmetry into this process to more precisely regulate the flow of semantic information across steps. In our design, **identity visual semantics** are injected exclusively during the zig (exploration) step to establish strong identity grounding early in the denoising trajectory. The zag step then adjusts the latent space without further visual input, helping to **avoid overfitting to the identity**. Finally, the generation step focuses

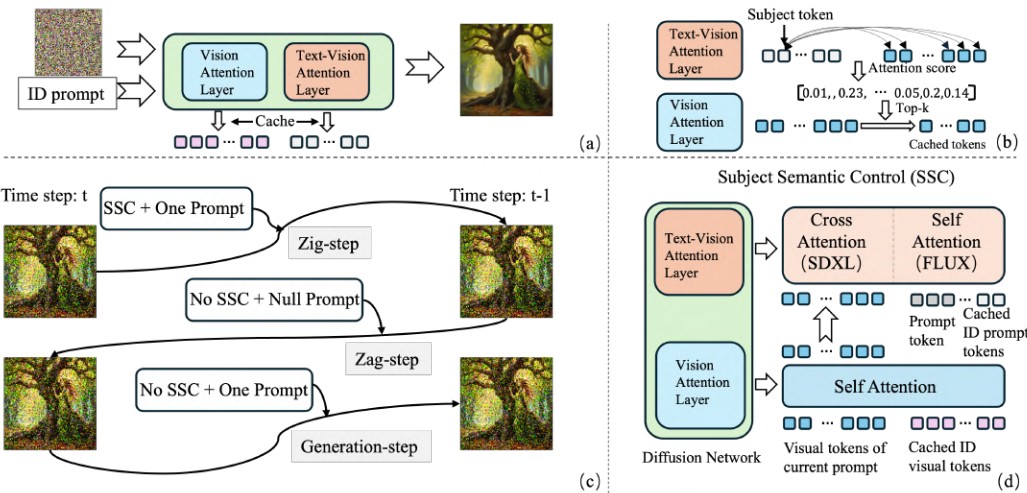

Figure 3: Overview of our proposed pipeline. (a) Identity-guided diffusion inference, where identity prompts are used to cache identity-related visual tokens. (b) Visual token selection module, which leverages attention scores to identify the most relevant tokens for the subject. (c) Illustration of the asymmetric design applied to zigzag sampling. (d) Integration of identity-aware visual information during the zigzag sampling process.

solely on **prompt alignment**, free from additional visual conditioning. This asymmetric configuration allows our method to achieve a superior balance between subject consistency and prompt fidelity, particularly in complex, multi-image story generation tasks. An overview of the proposed method is illustrated in Figure 3, with detailed step-by-step operations described below. Formally, let $x_t$ denote the noisy latent at timestep $t$, and $x_{t-1}$ the denoised latent. While the zig and generation steps follow the standard forward denoising trajectory, the zag step performs an inverse denoising operation, mapping $x_t$ to a prior latent $x_{t+1}$. The computation for each sub-step is governed by the noise schedule $\alpha_t$ predefined by the underlying diffusion model.

$$x_{t-1} = \sqrt{\alpha_{t-1}}\frac{x_t - \sqrt{1-\alpha_t}\epsilon_\theta(x_t)}{\sqrt{\alpha_t}} + \sqrt{1-\alpha_{t-1}}\epsilon_\theta(x_t) \qquad \hat{x}_t = \sqrt{\frac{\alpha_t}{\alpha_{t-1}}}x_{t-1} + \sqrt{\alpha_t}(\sqrt{\frac{1}{\alpha_t}-1} - \sqrt{\frac{1}{\alpha_{t-1}}-1})\epsilon_\theta(x_{t-1})$$

Diffusion denoising step          Inverse diffusion denoising step

Building on the bidirectional nature of zigzag sampling, which enables more effective semantic control during generation, we propose asymmetric zigzag sampling. In our approach, we inject strong subject-specific semantic cues into the intermediate network representation during the zig step, then propagate this information into the noisy latent space through the zag step, while keeping the generation step unchanged. This asymmetric design allows to enhance subject consistency across generated images without compromising the model's ability to accurately follow the textual prompt. We describe our asymmetric setup for zigzag sampling below, and summarize the proposed method in Algorithms 1 and 2 in the appendix.

**Asymmetry Visual Guidance.** As described above, we inject strong subject-specific cues into the latent representation during the *zig* step. To extract these cues, we first run the diffusion denoising process using an identity-focused prompt that describes only the visual characteristics of the subject. Following prior work [28], we compute text-image attention scores to identify subject-relevant visual tokens at each layer and timestep. These tokens serve as the basis for semantic injection, helping the model preserve subject identity across generated images. We denote the subject-related attention scores in the text-image attention layer as:

$$\mathcal{M}_{\text{subject}} = [s_1^{l,m}, s_2^{l,m}, \cdots, s_n^{l,m}],$$

where $l$ and $m$ denote the layer and timestep indices, respectively. Based on these scores, we cache the top-$k$ key and value projections of visual tokens from the image self-attention layers. This caching process is illustrated in Figure 3 (a) and the selected tokens are defined as:

$$I_{\text{key}}^{l,m} = [i_{\text{key},1}^{l,m}, i_{\text{key},2}^{l,m}, \cdots, i_{\text{key},k}^{l,m}], \quad I_{\text{value}}^{l,m} = [i_{\text{value},1}^{l,m}, i_{\text{value},2}^{l,m}, \cdots, i_{\text{value},k}^{l,m}].$$

During image generation for each prompt, these cached subject tokens are concatenated with the current visual tokens in the image self-attention layers at each layer and timestep. This integration is formalized as follows:

$$K_+^{i,l} = \text{Concatenate}(I_{\text{key}}^{l,m}, K^{i,l}) \qquad\qquad V_+^{i,l} = \text{Concatenate}(I_{\text{value}}^{l,m}, V^{i,l})$$

The query tokens remain unchanged. The updated key $K_+^{i,l}$, value $V_+^{i,l}$, and original query tokens are then used to perform standard self-attention, which updates the visual representation in the image attention layer. To avoid compromising the model's ability to follow text prompts, this semantic injection is applied only during the *zig* step. The *zag* and *generation* steps remain unchanged. This process is further illustrated in the Subject Semantic Control (SSC) module shown in Figure 3.

**Asymmetry Prompt Guidance.** For text-based guidance, we have first adapted an approach proposed in [29], which concatenates all scene-level prompts into a single sentence and dynamically reweights the contribution of each sub-sentence during the generation of the corresponding image. This allows the model to maintain narrative coherence across scenes while emphasizing the relevant textual content at each generation step. In addition, we have integrated the Identity-Preserving Contrastive Alignment (IPCA) technique [29] to further strengthen subject representation through the text prompt, reinforcing the model's ability to retain subject identity throughout the story. However, directly applying these text-guidance strategies within the standard zigzag sampling framework has led to suboptimal results. We hypothesize that the strong text-driven supervision during the *zag* step conflicts with the subject-specific information introduced through visual token injection, potentially overriding or distorting it. This interference degrades subject consistency and overall image quality. To address this issue, we introduce a novel **asymmetric zigzag prompt guidance** strategy. Specifically, we apply the enhanced text-guidance mechanisms only during the *zig* and *generation* steps, where they align well with semantic injection and image refinement. During the *zag* step, we instead use a null prompt to prevent conflicting and allow the subject-specific visual information to propagate unimpeded. This asymmetry helps preserve both textual relevance and subject consistency across the generated sequence.

# 4 Experiments

**Comparison Methods.** We integrate our method in two backbone architectures: the UNet-based SDXL [30] and the transformer-based FLUX [31] models. For the SDXL backbone, we compare our approach against several state-of-the-art baselines, including both training-based methods—The Chosen One [53], PhotoMaker [43], and IP-Adapter [24]—and training-free methods—ConsiStory [28], StoryDiffusion [27], and 1Prompt1Story [29]. For the FLUX model, as no prior methods have been demonstrated to support this architecture, we re-implement the most recent and competitive training-free method, 1Prompt1Story, on top of FLUX. We then use this as a baseline for comparison against our method. This dual-platform evaluation demonstrates the adaptability and effectiveness of our approach across diverse diffusion architectures.

**Evaluation Metrics.** Following prior work [29, 28, 27], we evaluate all models along two key dimensions: **prompt alignment** and **subject consistency**. To assess prompt alignment, we use the CLIP image and text encoders to compute the average CLIP-Score [54] between each generated image and its corresponding prompt. This metric reflects how well the visual content of the generated image aligns with the intended textual description. For subject consistency, we adopt two complementary metrics from previous studies [29]: DreamSim [55] and CLIP-I [54]. DreamSim measures perceptual similarity and has shown strong correlation with human judgment in evaluating visual coherence. CLIP-I computes the average cosine similarity between CLIP image embeddings, capturing the consistency of subject identity across different images. To ensure that subject consistency is not influenced by variations in background content, we follow prior evaluation protocols and apply CarveKit [56] to remove the background from each generated image. The removed regions are then filled with random noise, isolating the subject and allowing for a more focused and accurate evaluation.

# 5  Results

**Quantitative Comparison.** Table 1 presents a quantitative comparison between our proposed method and previously discussed approaches. As shown, our Asymmetry ZigZag Sampling technique achieves the best overall performance across all evaluation metrics among training-free visual storytelling methods. When compared with training-based methods, our approach outperforms both PhotoMaker and The Chosen One across all evaluation metrics. While the IP-Adapter method demonstrates the highest performance in DreamSim and CLIP-I scores, our method achieves a comparable CLIP-I score and significantly narrows the gap in DreamSim performance between training-free and training-based methods reducing it by 63.27%. Although the IP-Adapter excels in subject consistency metrics (CLIP-I and DreamSim), it performs considerably worse in text-alignment metrics such as CLIP-T. This discrepancy may be due to the IP-Adapter's tendency to overemphasize the subject, often generating images with highly similar or repetitive layouts, as illustrated in the figure 4.

| Method | Base Model | Train-Free | CLIP-T↑ | CLIP-I↑ | DreamSim↓ | Steps |
|---|---|---|---|---|---|---|
| Vanilla SDXL | - | - | 0.9074 | 0.8165 | 0.5292 | 50 |
| Vanilla FLUX | - | - | 0.8977 | 0.8494 | 0.3888 | 28 |
| The Chosen One | SDXL | ✗ | 0.7614 | 0.7831 | 0.4929 | 35 |
| PhotoMaker | SDXL | ✗ | 0.8651 | 0.8465 | 0.3996 | 50 |
| IP-Adapter | SDXL | ✗ | 0.8458 | **0.9429** | **0.1462** | 30 |
| ConsiStory | SDXL | ✓ | 0.8769 | 0.8737 | 0.3188 | 50 |
| StoryDiffusion | SDXL | ✓ | 0.8877 | 0.8755 | 0.3212 | 50 |
| 1Prompt1Story | SDXL | ✓ | 0.8942 | 0.9117 | 0.1993 | 50 |
| Ours | SDXL | ✓ | **0.8946** | 0.923 | 0.1798 | 50 |
| 1Prompt1Story | FLUX | ✓ | 0.8716 | 0.9118 | 0.1957 | 28 |
| Ours | FLUX | ✓ | **0.8949** | **0.9216** | **0.1843** | 28 |

Table 1: Quantitative Comparison. We report quantitative results for different methods. For the SDXL architecture, the best-performing value is highlighted in bold, while the second-best is indicated with a box. For the FLUX model, only the best result is highlighted in bold. The baseline models—vanilla SDXL and vanilla FLUX—are included as references but are excluded from the comparative ranking.

**User Study.** While automatic evaluation metrics offer a useful quantitative assessment, they can be biased due to their reliance on pretrained models. To further validate the effectiveness of our proposed method, we conducted a human evaluation study. We randomly selected 30 prompts from the benchmark dataset and generated corresponding image sequences using all competing methods. Twenty participants were invited for the user study. For each participant, a custom program randomly selected 20 out of the 30 prompts, and presented four resulting image sequences obtained with different methods for each selected prompt. Participants were asked to choose their preferred image sequence based on three criteria: identity consistency, prompt alignment, and overall image quality. As shown in Table 2, images generated by our method received the highest preference from participants, indicating strong alignment with human judgment. Further details of the user study protocol and interface can be found in the Appendix.

| Method | IP-Adapter | ConsiStory | StoryDiffusion | 1Pormpt1Story | Ours |
|---|---|---|---|---|---|
| **Percent (%)↑** | 5.15 | 19.18 | 16.25 | 24.50 | **35.02** |

Table 2: Human preference comparison among different methods.

**Qualitative Comparison.** Figures 4 and 6 present qualitative comparisons of our method against existing approaches. For the SDXL model, we compare our method with IP-Adapter [24], ConsisTory [28], StoryDiffusion [27], and 1Prompt1Story [29]. For the FLUX model, we evaluate our method against the training-free, state-of-the-art 1Prompt1Story approach. As shown in Figure 4, IP-Adapter tends to overemphasize identity consistency at the expense of faithfully representing the input text. Meanwhile, ConsisTory, StoryDiffusion, and 1Prompt1Story struggle to maintain subject

consistency across different images. In contrast, our method achieves a well-balanced performance, effectively preserving subject identity while accurately following the text descriptions. Figure 6 further underscores the superiority of our approach over the 1Prompt1Story model in terms of both subject consistency and text alignment. Additionally, it highlights the generalizability of our method, which maintains strong performance across diverse network architectures.

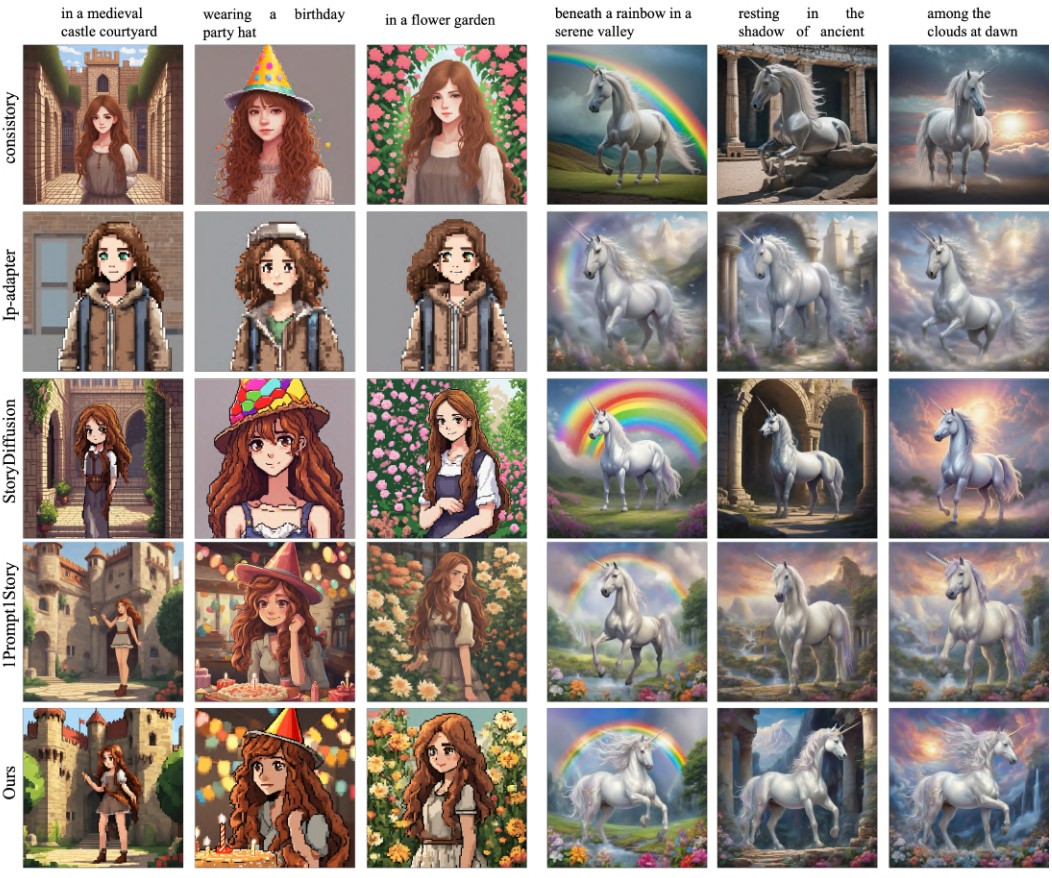

Figure 4: Qualitative Results Using the SDXL Backbone. We compare our method with four baselines: ConsisStory, IP-Adapter, StoryDiffusion, and 1Prompt1Story. The identity prompt is shown at the bottom, while individual image prompts are displayed above each corresponding image. Our method demonstrates a strong balance between maintaining subject consistency and adhering to textual prompts. In contrast, the baseline methods often struggle—either failing to preserve the subject's identity or deviating from the given text descriptions.

| Method | CLIP-T↑ | CLIP-I↑ | DreamSim↓ |
|---|---|---|---|
| Asymmetry | 0.8946 | 0.923 | 0.1798 |
| Zig-gen | 0.8534 | 0.919 | 0.1801 |
| Zig-zag | 0.8944 | 0.8916 | 0.2121 |
| All | 0.8788 | 0.8833 | 0.2312 |

Table 3: Quantitative comparisons of different zigzag sampling designs.

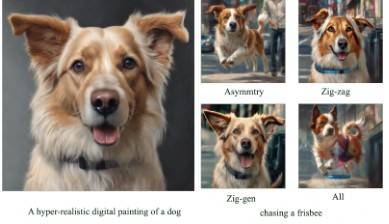

Figure 5: Qualitative comparisons of different zigzag sampling designs.

**Ablation study:Effect of Asymmetric Visual Injection**. We conduct experiments to evaluate the effectiveness of our proposed asymmetric visual injection design by comparing it with three alternative visual injection strategies: (1) zig-gen symmetric sampling, where visual semantics are injected during

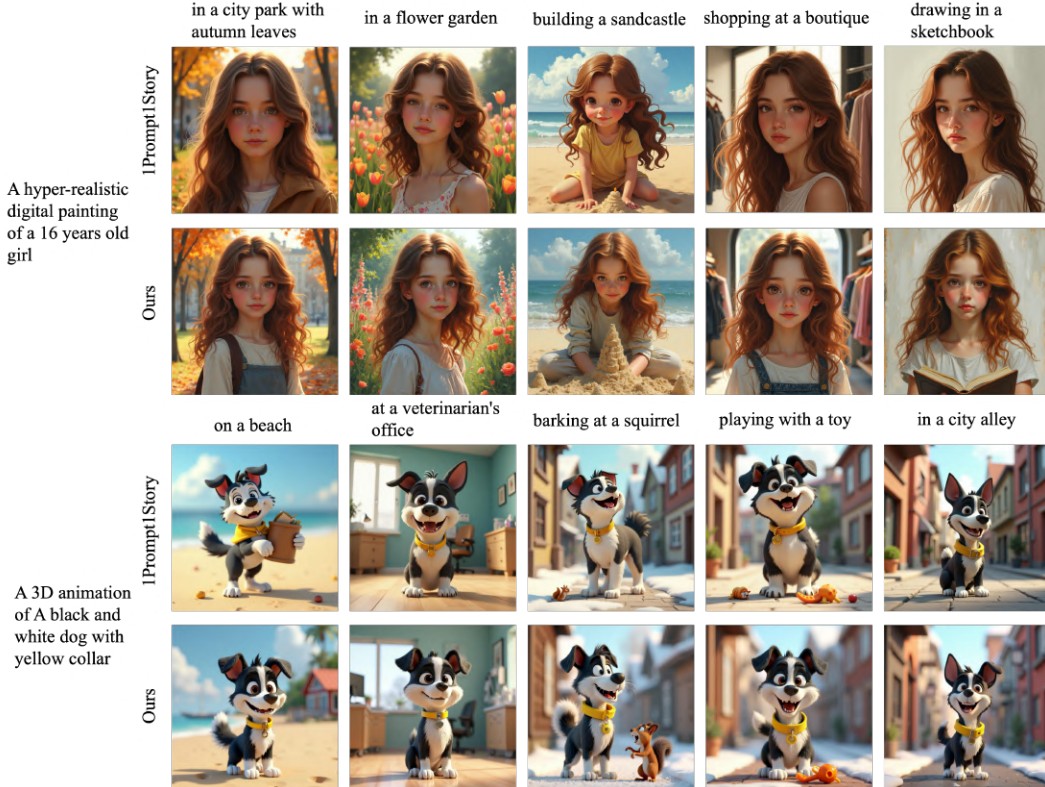

Figure 6: Qualitative Results: We compare our method with 1Prompt1Story using the FLUX backbone. Across varying prompts, our method demonstrates superior ability to preserve subject identity, highlighting its robustness in maintaining consistency.

the zig and generation steps but omitted in the zag step; (2) zig-zag symmetric sampling, which injects visual information in the zig and zag steps while excluding it during generation; and (3) fully symmetric sampling, where visual injection is applied across all three steps—zig, zag, and generation. These comparisons allow us to isolate the impact of asymmetric conditioning on balancing subject identity consistency and text-image alignment. Table 3 shows that fully symmetric sampling performs worst, likely due to over-conditioning, which introduces visual artifacts also observable in Figure 5. While the zig-gen strategy achieves higher identity similarity than zig-zag—likely due to visual injection during generation—it suffers from poor text alignment. Conversely, zig-zag provides better alignment at the expense of identity consistency. Our asymmetric visual conditioning design achieves the best overall balance between these two competing objectives, i.e., identity preservation and accurate adherence to textual prompts. These results validate the effectiveness of the proposed asymmetric strategy.

**Ablation Study: Effect of Asymmetric Prompt Conditioning**. In our method, we employ prompt conditioning in the zig and generation steps, while using a null (empty) prompt in the zag step. This asymmetric design is motivated by the intuition that removing the prompt in the zag step helps prevent overfitting to the text condition and

|  | CLIP-T↑ | CLIP-I↑ | DreamSim↓ |
|---|---|---|---|
| Symmetry | 0.9098 | 0.8841 | 0.2251 |
| Asymmetry | 0.923 | 0.8946 | 0.1798 |

Table 4: Quantitative comparisons of symmetry (using prompt condition in zag step) and asymmetry (using null prompt conditionin zag step) design.

allows the model to better retain the subject-specific information introduced during the zig step. To validate the effectiveness of this design, we compare a symmetric variant—where prompt conditioning is applied uniformly across all steps—with our asymmetric configuration. As shown in Table 4, the asymmetric design consistently achieves superior performance, with CLIP-T and CLIP-I scores improving from 0.9098/0.8841 to 0.923/0.8946, and DreamSim (lower is better) decreasing from 0.2251 to 0.1789. These results confirm that the asymmetric prompt conditioning facilitates stronger

semantic alignment with the textual prompt while preserving visual fidelity and preventing excessive prompt dependency, making it a more effective and balanced design choice for our framework.

**Ablation Study: The Influence of Hyperparameter** $k$ In our visual injection module, we select the top-$k$ visual tokens based on attention scores for injection during the sampling process. We evaluate the impact of varying $k$ values (ranging from $0.2$ to $0.8$) on generation performance (Table 5). The results show that increasing $k$ generally improves image similarity but slightly reduces text alignment. Notably, a value of $k = 0.2$ achieves the best balance between these two objectives. Although different $k$ values affect both metrics, the variations are minor, suggesting that our method is robust to the choice of $k$.

| k | CLIP-T↑ | CLIP-I↑ | DreamSim↓ |
|-----|---------|---------|-----------|
| 0.2 | 0.8946 | 0.923 | 0.1798 |
| 0.4 | 0.8931 | 0.9245 | 0.1799 |
| 0.6 | 0.8921 | 0.9258 | 0.1801 |
| 0.8 | 0.8924 | 0.9287 | 0.1827 |

Table 5: The influence of hyperparameter $k$ on the generation performance.

**Inference Time Comparison.** Our method can also be regarded as a form of test-time scaling in diffusion models. Compared with the standard diffusion inference process, our approach decomposes each inference step into three sub-steps—zig, zag, and generation—which inevitably introduces additional computational overhead and leads to longer inference time. To examine this effect, we evaluated the inference speed of our method against the regular diffusion process using both a UNet-based SDXL model and a transformer-based FLUX model. For SDXL with $50$ sampling steps, the baseline model required $7.44$ seconds per image, while our method took $21.33$ seconds. Similarly, for FLUX with $28$ sampling steps, the baseline achieved $1.50$ seconds per image, whereas our method required $5.50$ seconds. These results indicate that our approach increases the inference time by approximately $2.9$ on SDXL and $3.7$ on FLUX. Despite the additional computational cost, the improved subject consistency across images in visual storytelling justifies the longer inference time as a reasonable trade-off for enhanced generative performance.

**Other Applications.** Similar to the One-Prompt-One-Story approach [29], our method can be naturally extended to generate long stories containing an arbitrary number of images. This is achieved by applying a sliding-window strategy over the sequence of prompts, where each window of prompts is processed using our proposed method to ensure coherent subject and style consistency across adjacent images. By iteratively moving the window through the entire prompt sequence, our approach can maintain both local and global narrative consistency, enabling the generation of super-long visual stories. A visualization of such an extended story is provided in the appendix, demonstrating the capability of our method to handle narratives of considerable length while preserving high-quality and consistent image generation.

## 6    Conclusion

We propose an asymmetric zigzag sampling strategy to address the challenge of training-free visual storytelling, achieving a strong balance between subject consistency and text alignment. By leveraging the distinct roles of zig (exploration), zag (adjustment), and generation, our method injects visual semantics only where needed, enabling coherent, identity-consistent outputs without retraining. This approach also highlights the potential of sampling strategies as a promising direction for future research in controllable image generation.

## 7    Limitations

While our proposed method demonstrates strong performance in maintaining subject consistency and adhering to textual descriptions in visual story generation, it is not without limitations. First, the use of zigzag sampling—comprising three sub-steps per generation step—introduces additional computational overhead, which may increase inference time compared to standard sampling strategies. Second, although our approach is compatible with both the SDXL and FLUX architectures, our experiments indicate that it integrates more seamlessly with the FLUX model. In some cases, we observe occasional failures or reduced performance when applied to SDXL, suggesting that further architecture-specific optimization could enhance robustness and generalizability. Nonetheless, these limitations do not detract from the overall effectiveness of our method and highlight promising directions for future improvement.

# A  Acknowledgement

This work is funded by the CALCULUS project (European Research Council Advanced Grant H2020-ERC-2017-ADG 788506) and the Flanders AI Research Program.

# B  Boarder Impacts

Visual generative techniques, particularly text-to-image (T2I) models, hold significant potential for producing coherent visual content across diverse scenarios, making them highly applicable to downstream tasks such as storytelling and personalized content creation. One of the most challenging aspects in this domain is the consistent synthesis of characters across varying contexts—a problem that existing methods continue to struggle with, as discussed in this paper. Our proposed approach addresses this challenge by effectively balancing subject consistency and prompt fidelity, allowing users to generate coherent story sequences featuring the same character while closely adhering to their provided descriptions. In addition, our exploration of the unique structure of zigzag sampling introduces a new perspective on its utility in diffusion-based generation, offering valuable insights that may inspire future research into more controllable and semantically aware generative models.

The application of text-to-image models in visual storytelling, while creatively empowering, introduces significant ethical, privacy, and security risks. A major concern is the non-consensual creation of fictional narratives featuring real individuals, where realistic and consistent character generation enables the fabrication of defamatory, misleading, or harmful visual stories—such as fake memoirs, satirical comics, or illustrated scenarios depicting private citizens or public figures in compromising situations—without their knowledge or consent. The very capability to maintain character coherence across scenes, which enhances narrative immersion, can be exploited to produce persuasive, long-form synthetic content that blurs the line between fiction and reality, facilitating disinformation campaigns or emotional manipulation. Furthermore, privacy is compromised when models trained on unconsented web-scraped data generate characters closely resembling real people, effectively creating digital doppelgängers embedded in fictional universes. These capabilities also pose security threats, as coherent AI-generated visual stories can be weaponized for influence operations, identity exploitation, or viral misinformation, undermining trust and personal autonomy. Without robust safeguards—such as provenance tracking, consent filters, and transparent content policies—AI-driven visual storytelling risks enabling large-scale narrative abuse with profound societal consequences.

# C  Usage of LLMS

We only use large language models (LLMs) for writing assistance, such as correcting grammar, fixing typos, and improving clarity.

# D  Implementation Details

We implement our method on two open-source models: SDXL and FLUX. For SDXL, we use the *stabilityai/stable-diffusion-xl-base-1.0* version, and for FLUX, we adopt the *black-forest-labs/FLUX.1-dev* version. All baseline methods—including IP-Adapter [24], Consistory [28], StoryDiffusion [27], and 1Prompt1Story [29]—are reproduced using their official open-source implementations with default hyperparameters. Since IP-Adapter is designed for image-conditional generation, we adapt it to our setting by first generating an identity image using SDXL with the given identity prompt. This generated image is then used as the conditioning input for IP-Adapter to produce images guided by different prompts.

For the implementation of our method on the SDXL model, we cache visual tokens only from the mid and upper layers across all steps. Accordingly, feature injection during the zig step is also limited to these layers. We use a classifier-free guidance scale of $5.5$ for both the zig and generation steps, and set it to $0$ during the zag step. All experiments are conducted on a single NVIDIA A100 GPU.

For the FLUX model, which differs architecturally from SDXL by separating text-image and image-image interaction stages, we adopt a different strategy. FLUX begins with several layers of text-image interaction, followed by layers of purely image-based interaction. To cache visual tokens, we first average the attention scores from the early text-image interaction layers to identify subject-relevant

visual features. These selected tokens are then used for feature injection across all image-image interaction layers. Experiments for FLUX are also run on a single NVIDIA A100 GPU.

---

**Algorithm 1** Identity Visual Token Cache (Subject token extraction & top-$k$ selection)

---

**Require:** identity prompt $P_{\text{id}}$, pretrained diffusion model $\epsilon_\theta$, text encoder $\tau_\zeta$, time steps $\{m\}$ (used for identity extraction), layers $\mathcal{L}$, top-$k$ ratio $k$
**Ensure:** cached key tokens $\mathcal{I}^{\text{key}} = \{I^{\ell,m}_{\text{key}}\}$ and value tokens $\mathcal{I}^{\text{value}} = \{I^{\ell,m}_{\text{value}}\}$
 1: Compute text embedding $T_{\text{id}} \leftarrow \tau_\zeta(P_{\text{id}})$.
 2: **for** each layer $\ell \in \mathcal{L}$ and timestep $m$ used for identity extraction **do**
 3:   Run denoising pass (or inference pass) with prompt $P_{\text{id}}$ to obtain intermediate visual tokens at $(\ell, m)$.
 4:   Compute text-image attention scores $S^{\ell,m} = \text{AttentionScores}(\text{visual tokens}, T_{\text{id}})$.
 5:   Identify top-$k$ indices by score: $J^{\ell,m} \leftarrow \text{TopKIndices}(S^{\ell,m}, k)$.
 6:   Extract corresponding key / value projections:

$$I^{\ell,m}_{\text{key}} \leftarrow \{i^{\ell,m}_{\text{key},j} : j \in J^{\ell,m}\}, \quad I^{\ell,m}_{\text{value}} \leftarrow \{i^{\ell,m}_{\text{value},j} : j \in J^{\ell,m}\}.$$

 7:   Store $I^{\ell,m}_{\text{key}}$ into $\mathcal{I}^{\text{key}}$ and $I^{\ell,m}_{\text{value}}$ into $\mathcal{I}^{\text{value}}$.
 8: **end for**
 9: **return** $\mathcal{I}^{\text{key}}, \mathcal{I}^{\text{value}}$

---

---

**Algorithm 2** Asymmetric Zigzag Sampling with Zig Visual Sharing (ZVS) & Asymmetric Prompt Zigzag Inference (APZI)

---

**Require:** target prompt $P$, identity token caches $\mathcal{I}^{\text{key}}, \mathcal{I}^{\text{value}}$ (from Alg. 1), diffusion model $\epsilon_\theta$, text encoder $\tau_\zeta$, full zigzag time schedule $t = T, \ldots, 1$, and its noise coefficients $\alpha_t$
**Ensure:** latent $x_0$ (decoded to image by decoder $D$)
 1: Compute full prompt embedding $T \leftarrow \tau_\zeta(P)$ (can use one-prompt fusion / reweighting as in paper).
 2: Initialize noisy latent $x_T \sim \mathcal{N}(0, I)$.
 3: **for** each diffusion step index $t = T, T-1, \ldots, 1$ **do**
 4:   **Zig step (forward denoise + visual injection)**:
       1.   Compute standard denoising prediction $\hat{\epsilon} \leftarrow \epsilon_\theta(x_t, t, T)$ using prompt embedding $T$.
       2.   Perform denoising update (forward) to get intermediate latent $x_{t-1}$
       3.   **Inject identity visual tokens** into self-attention layers: for each layer $\ell$ used,

$$K^{\ell,+} \leftarrow \text{Concat}(I^{\ell,m}_{\text{key}}, K^\ell), \quad V^{\ell,+} \leftarrow \text{Concat}(I^{\ell,m}_{\text{value}}, V^\ell),$$

       where $K^\ell, V^\ell$ are the current layer key/value projections and $I^{\ell,m}_\cdot$ come from the caches. Keep queries unchanged. (Apply only in Zig.)
 5:   **Zag step (inverse denoise — null prompt / no text guidance)**:
       1.   Use null prompt.
       2.   Perform inverse denoising mapping to propagate the injected identity into the noisy latent:
$$\tilde{x}_t \leftarrow \text{InverseDenoise}(x_{t-1}, \ t-1 \to t, \ \epsilon_\theta, \ \text{null prompt}),$$
       3.   (No visual injection in Zag.)
 6:   **Generation step (final denoise with text guidance)**:
       1.   Use full prompt embedding $T$ to compute $\epsilon$-prediction on $\tilde{x}_t$.
       2.   Perform forward denoising to obtain $x^{\text{final}}_{t-1}$ (standard step).
       3.   Set $x_{t-1} \leftarrow x^{\text{final}}_{t-1}$ and continue.
 7: **end for**
 8: Decode $x_0$ to image: $I \leftarrow D(x_0)$ and return $I$.

---

# E   More Details about User Study

We conducted a user study to evaluate our method in comparison with four existing approaches: IP-Adapter [24], Consistory Model [28], Story Diffusion [27], and 1Prompt1Story [29]. All models were used to generate images based on prompts from the *ConsiStory+* benchmark, using the same random seeds as reported in their respective papers to ensure a fair comparison.

From the generated dataset, we randomly selected 30 prompts, each associated with 4 images. For each participant in the user study, the system randomly sampled 20 out of these 30 prompts to form an evaluation set. Before starting the evaluation, users were briefed on three key criteria:

- **Identity Consistency**: Measures whether the same character or subject appears consistently across all images for a given prompt.
- **Prompt Alignment**: Assesses how well each image reflects the content and intent of the original text prompt.
- **Image Quality**: Evaluates the overall visual quality, including clarity, detail, and aesthetic appeal.

To minimize potential bias, the presentation order of the five methods was randomized for each question in the study interface. Figure 10 presented the user interface of our user study system.

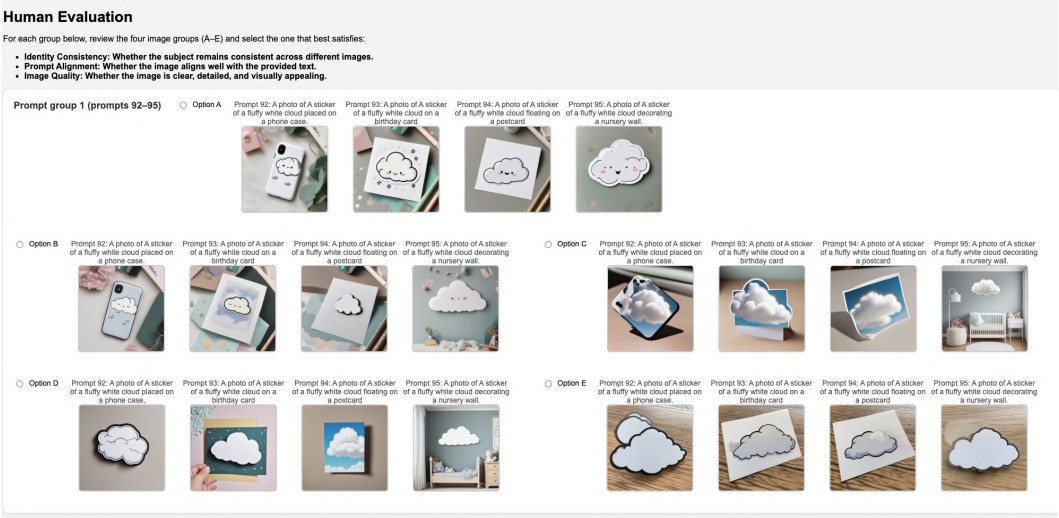

Figure 7: A visualization of the user interface used in our user study system.

# F   More Visualizations

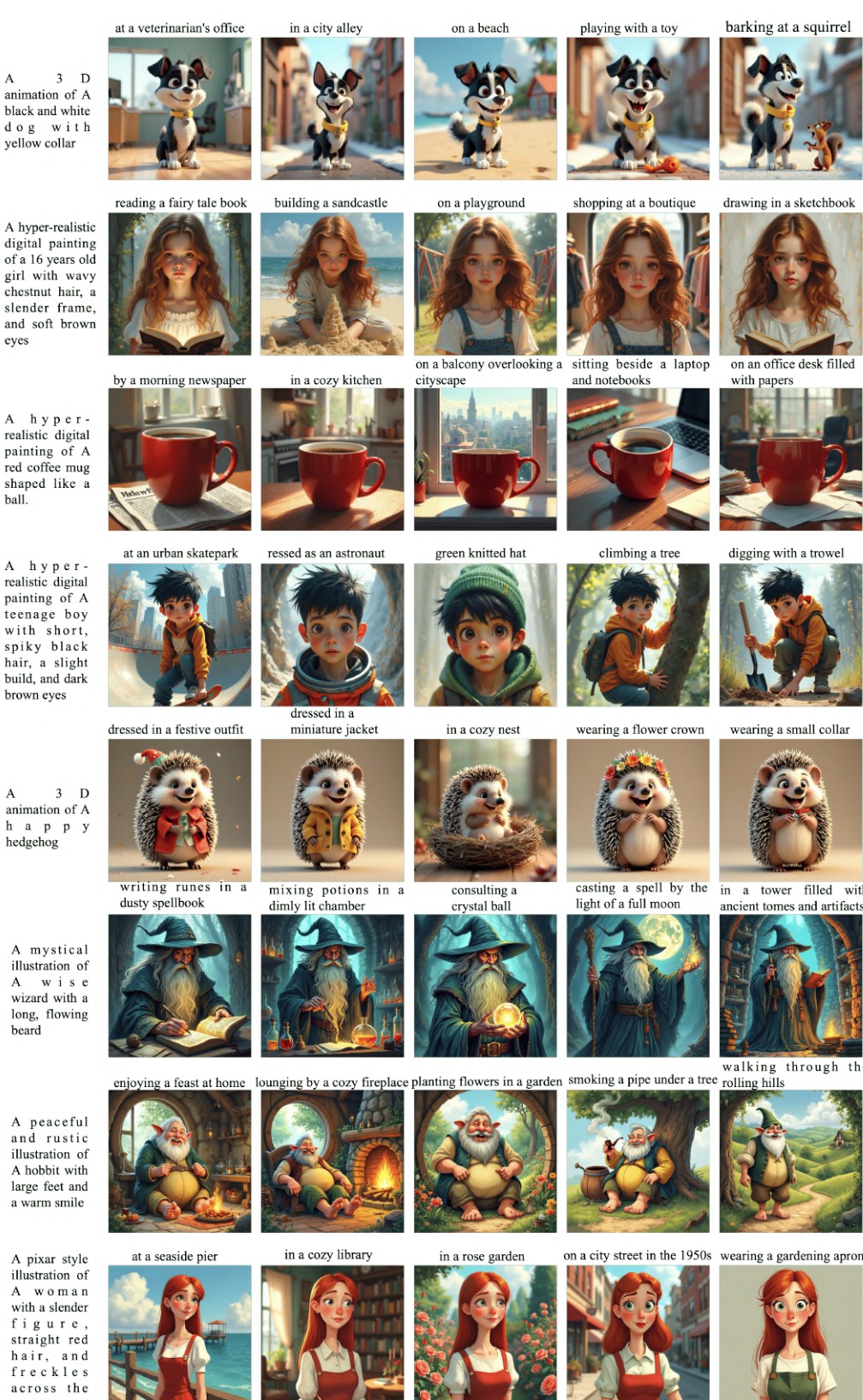

Figure 8: More visualzations of results generated using FLUX model.

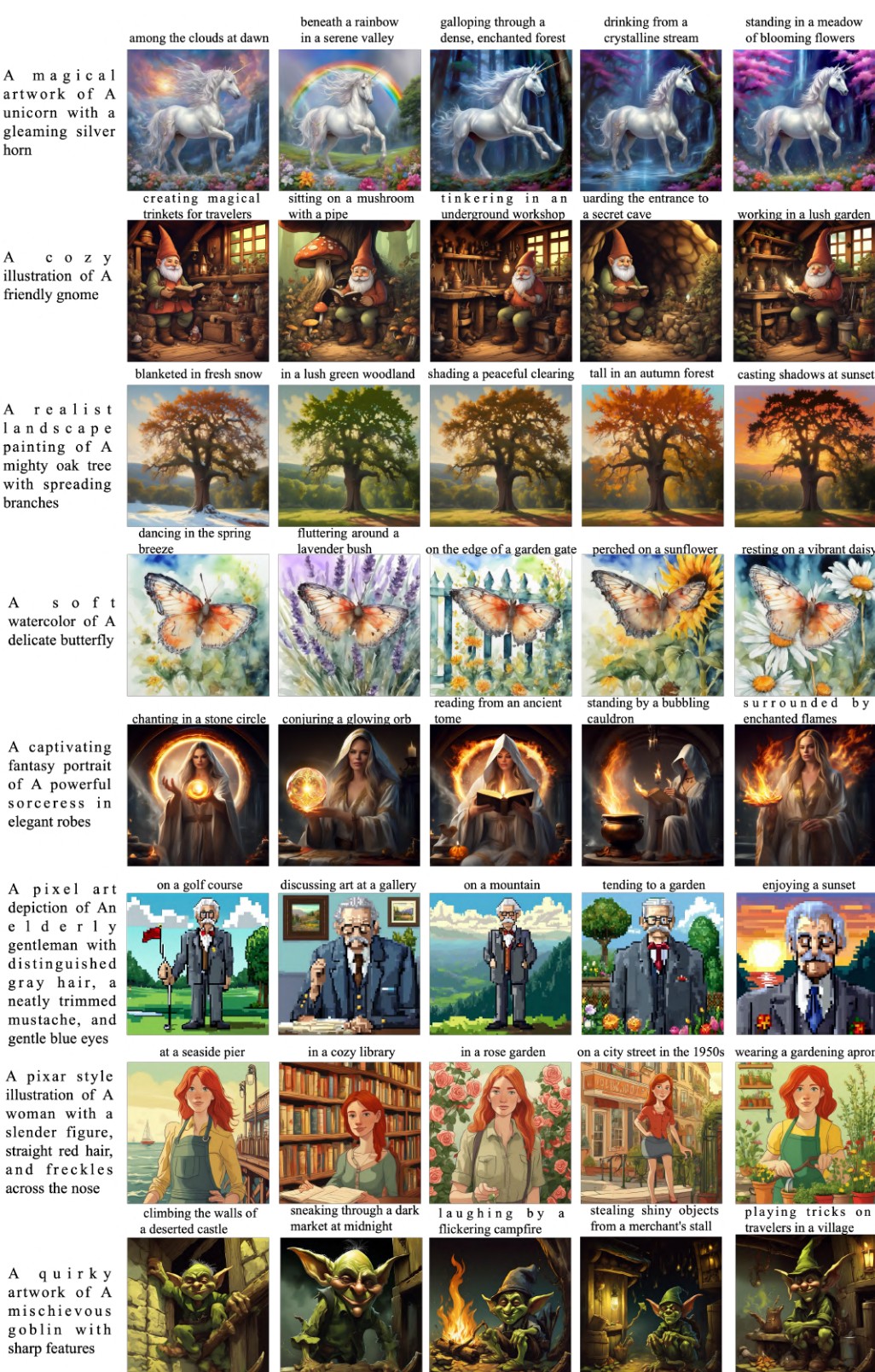

Figure 9: More visualzations of results generated using SDXL model.

# G  Long Story Visualizations

An anime-style illustration of a 16 years old girl with wavy chestnut hair, a slender frame, and soft brown eyes

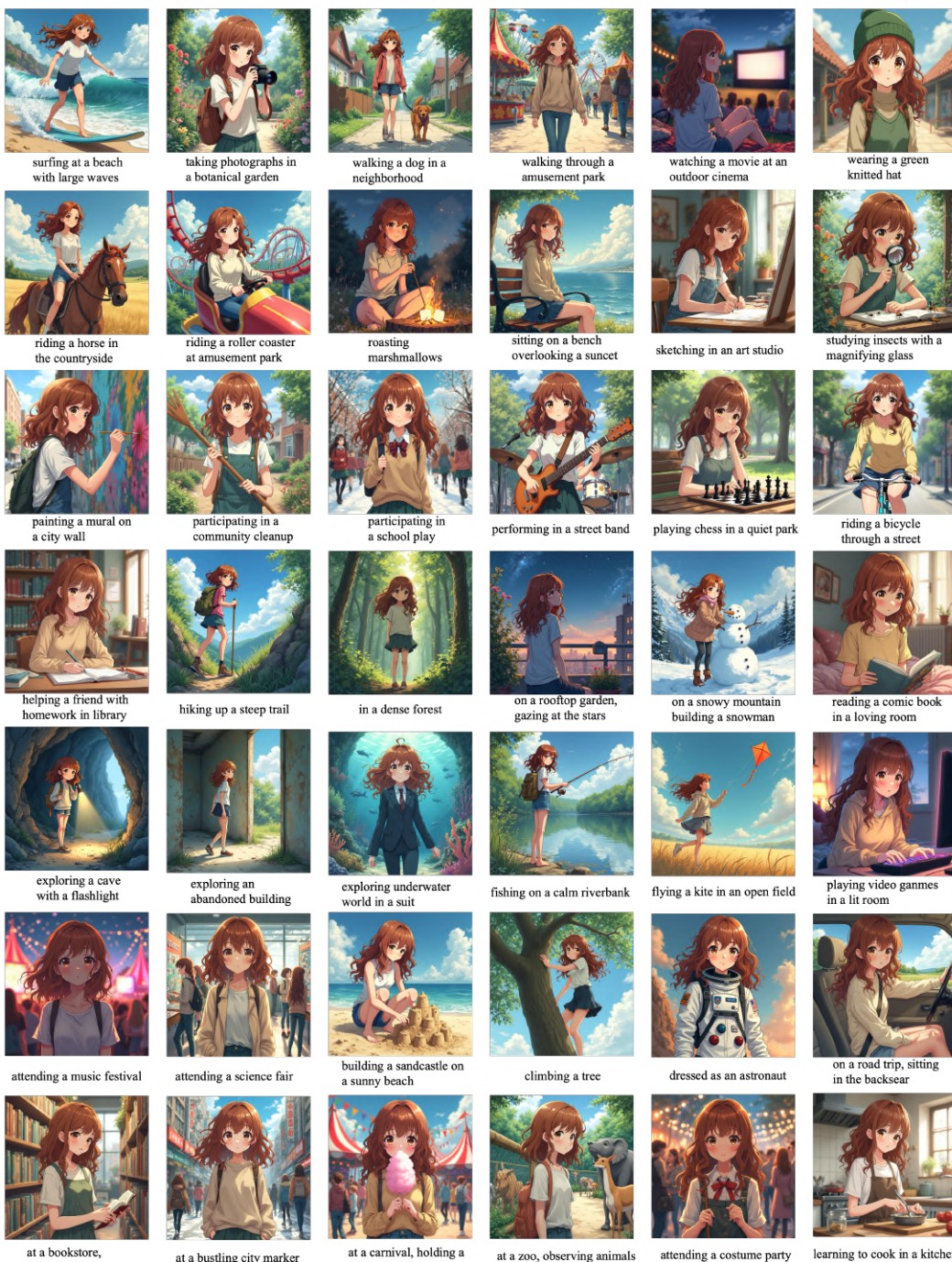

Figure 10: Long story generated using FLUX model

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
