# OpenReview forum: "Consistent Story Generation: Unlocking the Potential of Zigzag Sampling"
_NeurIPS.cc/2025/Conference — NeurIPS 2025 poster_

### Official Review · Reviewer_U3LM · 2025-07-01

**Clarity:** 3
**Significance:** 4
**Originality:** 3
**Rating:** 4
**Confidence:** 3

**Summary:**

This paper proposes Asymmetric Zigzag Sampling (AZS), a training-free strategy to enhance subject consistency in visual story generation. By decomposing each diffusion step into "zig-zag-generation" sub-steps, AZS integrates visual sharing and asymmetric prompting: the zig step injects subject-specific features via attention-based token caching, the zag step refines latents with null prompts to avoid overfitting, and the generation step ensures text alignment. Experiments on SDXL (UNet) and FLUX (Transformer) show AZS outperforms baselines in subject consistency (CLIP-I, DreamSim) and text fidelity (CLIP-T), with 35.02% user preference for generated sequences. Ablations validate the asymmetric design’s effectiveness and hyperparameter robustness, demonstrating broad applicability across architectures.

**Questions:**

Mentioned in the weaknesses. If my concerns can be addressed, I will increase my rating.

**Ethical Concerns:**

["NO or VERY MINOR ethics concerns only"]

**Final Justification:**

The authors have clearly addressed the raised issues and resolved most of my concerns. However, given the notable increase in inference cost, I still maintain my original rating of 4 (Borderline accept).

**Limitations:**

Yes.

**Paper Formatting Concerns:**

No.

**Quality:**

3

**Strengths And Weaknesses:**

**Strengths**
1. Innovative Method Design: The Asymmetric Zigzag Sampling (AZS) strategy is proposed, combining a zigzag sampling mechanism with a visual sharing module to enhance subject consistency in multi-image story generation without extensive fine-tuning. By decomposing the generation process into "zig-zag-generation" sub-steps, AZS balances subject consistency and text alignment via asymmetric prompting and visual feature transfer, offering a novel training-free approach for visual storytelling.
2. Comprehensive Experimental Validation: Experiments on two architectures (SDXL and FLUX) compare against various training-based and training-free baselines. Quantitative results show AZS significantly outperforms state-of-the-art methods in subject consistency metrics (CLIP-I, DreamSim) while maintaining high text alignment (CLIP-T). User studies further confirm that AZS-generated sequences are preferred for identity consistency and overall quality.
3. Ablation Studies and Generalization: Ablations validate the effectiveness of the asymmetric design (vs. symmetric sampling) and the robustness of hyperparameter k. The method’s success across UNet-based and Transformer-based architectures demonstrates strong generalization.

**Weaknesses**
1. Qualitative results fail to clearly demonstrate consistency advantages over baselines: For instance, in the image generated in Figure 4, the appearance and shape proportions of the sheep people also change significantly; The color and appearance of the horse have changed significantly. Annotate key subject traits to highlight consistent preservation and contrast with baselines’ inconsistencies.   If instability occurs, analyze causes (e.g., sampling noise) and propose mitigations.
2. Computational overhead lacks quantitative analysis: Decomposing each generation step into three sub-steps introduces additional computation, but the paper only mentions "computational cost" without metrics (e.g., inference time per image). This limits evaluation of its applicability to real-time scenarios.
3. Architectural adaptation analysis is insufficient: While AZS integrates more smoothly with FLUX than SDXL, the paper does not analyze why. For instance, can the feature injection strategy of SDXL be further optimized? This may limit the universality demonstration of the method on different models
4. Typography issues in Figures 8 and 9: Uneven letter spacing in left-side text causes readability problems, requiring formatting corrections.

---

> ### Author Rebuttal · Authors · 2025-07-30
>
> *W1. Qualitative results fail to clearly demonstrate consistency advantages over baselines: For instance, in the image generated in Figure 4, the appearance and shape proportions of the sheep people also change significantly; The color and appearance of the horse have changed significantly. Annotate key subject traits to highlight consistent preservation and contrast with baselines’ inconsistencies. If instability occurs, analyze causes (e.g., sampling noise) and propose mitigations.*
>
> [Response]
> We thank the reviewer for this critical observation. We agree that while our method significantly improves consistency, the inherent stochasticity of diffusion models can still lead to minor variations. The primary goal of our qualitative examples was to show a superior balance between subject consistency and prompt fidelity compared to baselines, which often fail dramatically on one of these two axes.
> However, we recognize that the presentation of these results can be substantially improved. To make our method's advantages clearer, we will implement the reviewer's excellent suggestion.
> **Annotated Figures:** We will revise Figure 4 (and Figures 6, 8, 9) to include annotations with call-out boxes and arrows. These annotations will explicitly highlight the key subject traits that are consistently preserved by our method (e.g., the satyr's specific horn shape and reed pipe, the centaur's bow and torso markings) and contrast them with the clear inconsistencies in the baseline results (e.g., changes in subject identity, failure to include key objects).
> **Analysis of Instability:** We will add a sentence to the qualitative comparison section acknowledging that minor variations can still occur. We will attribute this to the exploration phase of the zigzag sampling, which is essential for generating diverse scene compositions, and frame it as a deliberate trade-off between rigid identity replication and creative diversity. This will better contextualize the results.
>
> *W2. Computational overhead lacks quantitative analysis: Decomposing each generation step into three sub-steps introduces additional computation, but the paper only mentions "computational cost" without metrics (e.g., inference time per image). This limits evaluation of its applicability to real-time scenarios.*
>
> [Response]
>  Inference time comparison:  (seconds  per image).
> SDXL[50steps]  7.44s ,  Ours [50 steps] 21.33s.
> FLUX[28steps]  1.50s ,  Ours  [28 steps]   5.5 s
>
> *W3. Architectural adaptation analysis is insufficient: While AZS integrates more smoothly with FLUX than SDXL, the paper does not analyze why. For instance, can the feature injection strategy of SDXL be further optimized? This may limit the universality demonstration of the method on different models*
>
> [Response]
> This is an insightful question, and the architectural differences between FLUX and SDXL likely contribute to FLUX's better compatibility with our method.
> **FLUX's Architectural Advantages:** FLUX is a transformer-based model that utilizes a unified self-attention mechanism with modality-specific projection layers. This design allows FLUX to fuse semantic information across visual and textual modalities without relying on explicit cross-attention, leveraging the transformer's strength in modeling complex dependencies. In our method, this means that the newly injected subject visual tokens, text tokens, and current generated visual tokens are jointly considered within FLUX's unified attention layers. This simultaneous interaction likely enables FLUX to better balance subject consistency and prompt following, leading to improved performance.
> **SDXL's Architectural Considerations:** In contrast, SDXL is a U-Net-based model that combines convolutional and transformer blocks. Its transformer blocks contain separate cross-attention layers for text information and self-attention layers for visual information flow between patches. When we inject subject visual tokens into SDXL's self-attention layers, the interaction between these subject tokens and the text tokens (handled by separate cross-attention layers) might be less integrated. This separation could make it more challenging for SDXL to optimally balance text following and subject consistency, as the model needs to reconcile information from distinct interaction pathways.
>
> We believe there is still space to further improve the feature injection strategy in SDXL, and this is a promising direction for future work. Based on our analysis, we have identified several potential optimization strategies:
>
> **Prompt-Aware Dynamic Injection:** Currently, we cache a static set of top-k visual tokens representing the subject's identity. A more advanced strategy would be to dynamically select or weight which subject tokens to inject based on the current scene's prompt. For example, for a prompt like "a dog running in a field," tokens corresponding to the dog's body and legs would be prioritized. For "a close-up portrait of a dog," face and fur texture tokens would be more relevant. This would create a more direct and semantic link between the textual guidance and the visual injection.
> **Spatio-Temporal Adaptive Injection:** Our current method injects features into the mid and upper layers of the U-Net during each zig step. This could be optimized by making the injection adaptive:
>         - **Spatially (Layer-wise):** Inject high-level structural features (e.g., face shape) into deeper layers of the U-Net and more fine-grained texture details (e.g., fur pattern) into earlier layers.
>         - **Temporally (Step-wise):** Apply stronger visual injection during the initial diffusion timesteps to firmly establish the subject's identity, and then gradually reduce its strength in later steps to give more control to the text prompt for refining details and scene composition.
> **Cross-Attention Modulation:** A more sophisticated approach would be to create a more direct bridge between the visual identity and text guidance pathways. Instead of only injecting tokens into the self-attention layers, the cached subject features could be used to modulate the output of the cross-attention layers. This would guide the model to interpret the text prompt in the context of the desired subject's appearance, potentially resolving the architectural separation more effectively.
> We are excited by these possibilities and thank the reviewer for prompting this valuable discussion. We will add a paragraph outlining these future research directions to the conclusion or limitations section of our final manuscript to acknowledge this potential for improvement and guide future work in this area.
>
> *W4.Typography issues in Figures 8 and 9: Uneven letter spacing in left-side text causes readability problems, requiring formatting corrections.*
>
> [Response]
> We thank the reviewer for spotting this formatting error. This was an oversight in the script used to generate the figures.
> We will regenerate Figures 8 and 9 for the final manuscript, ensuring that all text is formatted with proper letter spacing and is fully legible.

---

### Official Review · Reviewer_yE2y · 2025-07-02

**Clarity:** 3
**Significance:** 4
**Originality:** 3
**Rating:** 4
**Confidence:** 4

**Summary:**

The paper introduces a training-free asymmetric zigzag scheme in text-guided diffusion to preserve subject-specific visual features across different text prompts. The zigzag scheme is designed specifically to optimize both adherence to ID preservation while still giving the diffusion process sufficient freedom to adhere to the text prompt. The scheme consists of three stages: a) During the zig step, strong subject-specific semantic cues are injected. b) This is propagated into the noisy latent by the zag step, which performs inverse denoising to get closer to the final output. c) Finally, the generation step is responsible for aligning with the text prompt. The resulting visual quality outperforms all existing approaches, striking a good balance between subject consistency and text guiding.  While others might even be stronger in ID preservation they typically suffer from adhering less to the prompt. The method has been integrated into two diffusion backbones - SDXL and Flux.

**Questions:**

Please try to provide a few more exact details on the individual steps,
- In which layers is the zigzag pattern applied?
- How much longer does it take compared to simple text-guided diffusion?
- How many subject-specific tokens are used?

Please discuss some of the noticeable ID deviations.

**Ethical Concerns:**

["NO or VERY MINOR ethics concerns only"]

**Final Justification:**

In the reviews and the rebuttal the authors clearly presented how they will update the text to improve clarity. I therefore, still suggest accepting the paper.

**Limitations:**

yes

**Paper Formatting Concerns:**

none.

**Quality:**

3

**Strengths And Weaknesses:**

Strength
- The authors present a training-free technique that strikes a good balance between subject consistency and text guidance.
- The method is thoroughly validated, clearly showing qualitative and quantitative improvements compared to other existing schemes.
- The proposed zigzag scheme might be applicable to other applications as well.

Weaknesses
- The writing is not always as clear as it should be.
     - Fig. 3 does not contain the labels zig,zag, and generation in the for sub-images.
     - In Sec.3.2 the zig, zag and generation steps are defined multiple times, each time emphasizing a different aspect.
     - The abbreviations used in the sketches and in the tables are not always introduced in the text or are just slightly different, e.g. Fig 3. "SSC", and particularly irritating in the ablation study (page 8)  "asymmetric" vs. "all"   or "fully symmetric", ...
- The paper does not provide all necessary details for re-implementation, e.g. how many subject tokens are extracted and induced. How much longer does the final inference really take with the zigzag scheme. Making the publically available would be helpful.
- The discussion on not-so-well-performing cases could be more detailed. In Fig 6. the faun loses its wings or changes from more female to male appearance. The rainbow, while consistently showing a white border sometimes loses its clouds or looks more plastic than paper.

A smaller note: In Figure 4, none of the approaches is actually depicting a centaur (half human-half horse).

---

> ### Author Rebuttal · Authors · 2025-07-30
>
> *W1.1 Fig. 3 does not contain the labels zig,zag, and generation in the for sub-images.*
>
> [Response]
> We will update Figure 3(c) to explicitly label the "zig," "zag," and "generation" sub-steps in the timeline, making the diagram a much clearer illustration of our method as described in the text.
>
> *W1.2 In Sec.3.2 the zig, zag and generation steps are defined multiple times, each time emphasizing a different aspect.*
>
> [Response]
> In Sec.3.2 We first introduce the foundational zigzag sampling mechanism, outlining the general role each step plays (e.g., zig for exploration, zag for refinement, generation for final output). Following this, we explain our motivation for introducing an asymmetric information flow within this pipeline to better balance subject consistency and prompt following. We will revise our current writing to improve clarity.
>
> *W1.3 Inconsistent Abbreviations*
>
> [Response]
> Thank you for pointing out this oversight. Consistency is key for clarity. We will explicitly define "SSC (Subject Semantic Control)" upon its first mention and ensure it is used consistently throughout the paper and in Figure 3(d).
> The different methods in Table 3 follows below definition (line 279-284):
> "Asymmetry" (Our Method): Visual injection (ZVS) only in zig step. Text guidance (APZI with full prompt) in zig and generation steps. Null prompt in zag step.
> "Zig-gen": Visual injection (ZVS) in zig and generation steps. Text guidance (full prompt) in zig and generation steps. Null prompt in zag step.
> "Zig-zag": Visual injection (ZVS) in zig and zag steps. Text guidance (full prompt) in zig and generation steps. Null prompt in zag step.
> "All" (Fully Symmetric): Visual injection (ZVS) in zig, zag, and generation steps. Text guidance (full prompt) in zig and generation steps. Null prompt in zag step.
> We will clarify this distinction in the revised paper, making it explicit that these ablations examine the impact of where visual conditioning is applied within our established asymmetric zigzag framework, not variations of the zigzag process itself.
>
> *W2  Missing Implementation Details.*
>
> [Response]
> Number of Subject Tokens (k): As shown in our ablation study (Table 4, page 9), we experimented with different values for the hyperparameter k. We found that setting k=0.2 (i.e., caching the top 20% of visual tokens based on attention scores) achieves the best balance between subject consistency and text alignment. We will highlight this specific value in our main Implementation Details section (Appendix C) to make it more prominent.
>
> Inference Time Overhead:
> Inference time comparison: (seconds per image).
> SDXL[50steps] 7.44s , Ours [50 steps] 21.33s.
> FLUX[28steps] 1.50s , Ours [28 steps] 5.5 s.
>
> Making code publicly available: We will include a commitment in the paper to release our source code and detailed instructions upon publication to facilitate reproducibility.
>
>
> *W3 Discussion of Failure Cases.*
>
> [Response]
> This is an excellent suggestion that will add valuable nuance to our paper. We will perform a more detailed analysis of these cases.
> We will add a new subsection in the Appendix titled "Analysis of Failure Modes and Limitations," which will be referenced from the main text. In this section, we will discuss:
> The Faun (Fig. 6): We will analyze how semantic changes in prompts (e.g., from an active "dancing" pose to a passive "resting" one) can sometimes lead to the model dropping secondary attributes (like the wings), which might not have been encoded as a core part of the subject's identity.
> The Rainbow (Fig. 6): We will discuss how our method successfully preserves the core identity ("sticker of a vibrant rainbow"), but can show minor inconsistencies in secondary elements (clouds) which is not specifically mentioned in the prompt. This is a known challenge in generative models, and we will frame our discussion in this context.
> This analysis will provide a more balanced perspective on the method's strengths and current boundaries.
>
> Minor Note on Figure 4 (Centaur):
> Thank you for pointing this out. We acknowledge that the difficulty in generating specific composite subjects, such as centaurs, appears to be an inherent limitation of the underlying text-to-image base models themselves, as we observed consistent challenges even when using the vanilla base model without our proposed method. We will revise Figure 4 to include a more representative example where the base model performs adequately, and we will add a short discussion in the limitations section or an appendix to address these inherent shortcomings of the base models in generating certain complex or hybrid subjects. This will clarify that such generation failures are not a specific limitation of our method but rather a challenge inherited from the foundational models.
>
> *Q1. In which layers is the zigzag pattern applied?*
>
> [Response]
> This is detailed in Appendix C (line 344-353) and will be made clearer. For SDXL, visual token caching and injection are applied only in the mid and upper layers of the U-Net. For FLUX, we cache tokens from the early text-image interaction layers and inject them into all subsequent layers.
>
> *Q2. How much longer does it take compared to simple text-guided diffusion?*
>
> [Response]
> See above responses to W2.
>
> *Q3. How many subject-specific tokens are used?*
>
> [Response]
> See above responses to W2.

---

> ### Comment · Reviewer_yE2y · 2025-08-05
>
> The authors, thank you for your elaboration and the clarifications. I will keep my slightly positive score.

---

### Official Review · Reviewer_4vAP · 2025-07-03

**Clarity:** 3
**Significance:** 2
**Originality:** 2
**Rating:** 4
**Confidence:** 4

**Summary:**

The paper introduces Asymmetry Zigzag Sampling (AZS), a novel training-free sampling strategy for enhancing subject consistency in visual story generation using text-to-image models. The authors address a key challenge in multi-scene generation: maintaining consistent subject identity across images while preserving alignment with textual prompts. The proposed method decomposes each generation step into three sub-steps (zig, zag, generation) and integrates two main components:
1. Zig Visual Sharing (ZVS): Injects subject-specific visual cues during the "zig" step to enforce consistency.
2. Asymmetric Prompt Zigzag Inference (APZI): Uses a full prompt in the zig/generation steps and a null prompt in the "zag" step to balance subject fidelity and text alignment.
The method is evaluated on SDXL and FLUX architectures, outperforming existing training-free and training-based baselines in both quantitative metrics (CLIP-T, CLIP-I, DreamSim) and human evaluations. Key contributions include the asymmetric sampling framework, robust cross-architecture performance, and insights into latent-space conditioning for controllable generation

**Questions:**

1. The method adds sub-steps per generation. Could the authors quantify the runtime overhead compared to baselines (e.g., steps/second)?
2. What is the intuition behind using a null prompt here? Does empirical evidence support this over other designs (e.g., weakened prompt)?
3. The top-k ablation (Table 4) shows minor metric variations. Are there failure cases where k significantly impacts output?
4. The paper notes occasional SDXL failures. Are there architectural insights explaining FLUX’s better compatibility?

**Ethical Concerns:**

["NO or VERY MINOR ethics concerns only"]

**Final Justification:**

In their reviews and rebuttal, the authors have clearly outlined their planned textual revisions to enhance clarity. These improvements adequately address the concerns raised, and I therefore maintain my recommendation to accept the paper.

**Limitations:**

The paper adequately discusses limitations (e.g., computational overhead, SDXL compatibility) in Appendix A. Suggestions for improvement:
1. Expand on potential misuse (e.g., deepfake stories) and mitigation strategies (e.g., watermarking).
2. While training-free, could the method handle longer story sequences (>10 images)? A stress test would be valuable.

**Paper Formatting Concerns:**

No major formatting issues found. The paper adheres to the NeurIPS 2025 formatting guidelines.

**Quality:**

3

**Strengths And Weaknesses:**

Strengths
1. Rigorous experiments validate the method’s superiority over baselines, including ablation studies and user evaluations. The dual-architecture (SDXL/FLUX) validation strengthens generalizability.
2. The paper is well-structured, with clear explanations of the zigzag mechanism and visual/prompt asymmetry. Figures (e.g., Figure 2) effectively illustrate the pipeline.
3.  Addresses a critical gap in training-free consistent story generation, offering a practical solution with lower computational costs than fine-tuning approaches.
4. The asymmetric design is novel, distinguishing AZS from prior symmetric feature-sharing methods (e.g., StoryDiffusion, ConsiStory).

Weaknesses
1. While computational overhead is acknowledged, deeper analysis of inference time vs. baselines would strengthen practicality claims.
2. The role of the "zag" step (inverse denoising) could be elaborated further—why is a null prompt optimal here?
3. The zigzag concept builds on prior work (e.g., [51]); the paper could better delineate its unique adaptation for asymmetry.

---

> ### Author Rebuttal · Authors · 2025-07-30
>
> *W1.While computational overhead is acknowledged, deeper analysis of inference time vs. baselines would strengthen practicality claims.*
>
> [Response]
> Inference time comparison:  (seconds  per image).
> SDXL[50steps]  7.44s ,  Ours [50 steps] 21.33s.
> FLUX[28steps]  1.50s ,  Ours  [28 steps]   5.5 s
>
> *W2. The role of the "zag" step (inverse denoising) could be elaborated further—why is a null prompt optimal here?*.
>
> [Response]
> The "zig" step is designed for exploration and initial injection of strong subject-specific visual information. The "zag" step, performing inverse denoising, acts as a refinement and propagation phase for the latent representation. Our intuition is that after injecting visual identity cues in the zig step, applying a strong textual prompt or further visual conditioning in the zag step would cause immediate overfitting to the scene's textual description or over-constrain the subject's appearance too early, potentially disrupting the delicate balance needed for flexible storytelling. A null prompt in the zag step allows the visually grounded subject information to propagate through the denoising process without being pulled strongly by any specific text or new visual cues. This enables the subject identity to "settle" into the latent representation in a more generalized manner, preparing it for the final "generation" step which then focuses purely on aligning with the detailed scene prompt. This prevents the model from overly focusing on the current scene's textual details for subject definition, thus promoting better consistency across different scenes while preserving diversity.
> We conduct an additional ablation study to empirically support the optimality of the null prompt in the zag step, as also suggested by the other reviewer.
> Asymmetry + Zag-Text": Our "Asymmetry" method, but with the full textual prompt also applied during the zag step, instead of a null prompt.
> Asymmetry + zag-text:              clip-I  0.9098, clip-T 0.8841, DreamSim 0.2251
> Asymmetry + zag-null (ours):     clip-I  0.923,   clip-T 0.8946, DreamSim 0.1789
> The results above show that adding text in zag step leads to significant decreases in subject consistent evaluation metrics including clip-I and DreamSim scores. This demonstrates that the null prompt is indeed optimal for preserving a consistent subject identity across the story. We will include these results and discussion in the revised paper.
>
> *W3. The zigzag concept builds on prior work (e.g., [51]); the paper could better delineate its unique adaptation for asymmetry.*
>
> [Response]
> We agree. While [51] (Golden Noise for Diffusion Models) explores the functional roles of zig (exploration), zag (refinement), and generation steps in general, our work uniquely adapts and leverages this structure for asymmetric information flow aimed specifically at the subject consistency challenge in visual storytelling.
> We will explicitly delineate our unique adaptation in the revised "Related Work" and "Method" sections:
> Prior Work ([51]): Introduced the zigzag decomposition and identified the general roles of zig (exploration), zag (refinement), and generation.
> Our Unique Adaptation: We capitalize on these identified roles by proposing a novel, asymmetric strategy for when and how to inject different types of semantic information:
> 1. **Zig (Exploration)**: Dedicated to initial, strong subject identity grounding via visual token injection (ZVS). This sets the consistent subject foundation.
> 2. **Zag (Refinement/Propagation)**: Utilizes a null prompt (APZI) to allow the subject identity to propagate and refine in the latent space without being constrained by scene-specific text or further visual conditioning, preventing overfitting.
> 3. **Generation (Final Alignment)**: Focuses solely on text alignment with the full narrative prompt, relying on the already-consistent subject from previous steps.
>
>  This precise, purposeful asymmetry of information injection (visual vs. textual, and null vs. full prompt) at specific stages of the zigzag process is our distinct contribution, directly addressing the core problem of balancing consistency and fidelity, which [51] did not explore.
>
> *Q1. The method adds sub-steps per generation. Could the authors quantify the runtime overhead compared to baselines (e.g., steps/second)?*
>
> [Response]
> See response to W1.
>
> *Q2. What is the intuition behind using a null prompt here? Does empirical evidence support this over other designs (e.g., weakened prompt)?*
>
> [Response]
> See response to W2.
>
> *Q3. The top-k ablation (Table 4) shows minor metric variations. Are there failure cases where k significantly impacts output?*
>
> [Response]
> While Table 4 indicates that our method is robust to the choice of 'k' within the tested range (0.2 to 0.8) and variations are minor, we can elaborate on potential edge cases and the rationale for our optimal 'k' selection:
>
> **Extremely Low 'k' (e.g., <0.1):** If 'k' is too small, only a very limited number of visual tokens would be selected, potentially leading to insufficient subject information being injected. This could manifest as a degradation in subject consistency, as the model might not have enough cues to maintain the identity across different scenes.
>
> **Extremely High 'k' (e.g., >0.9):** Conversely, if 'k' is too high, it would mean injecting a very large proportion of visual tokens, including those that might be irrelevant or background-related. This could lead to an "over-conditioning" effect, similar to what is observed in our "fully symmetric" sampling ablation, resulting in visual artifacts or reduced generative diversity. It could also force the model to overemphasize the visual input, potentially hindering its ability to follow the textual prompt faithfully (lower clip-T).
>
> **Empirical Sweet Spot:** Our chosen value of k=0.2 represents an empirical sweet spot that achieves the best balance between subject consistency and text alignment. This suggests that selecting a moderate number of the most relevant subject tokens is key, rather than an exhaustive or minimal set. We will clarify this in the revised paper to provide more comprehensive insights into the implications of 'k' selection.
>
> We will present qualitative examples illustrating these potential subtle impacts for extreme k values in the Appendix to provide a more comprehensive understanding beyond just the quantitative metrics.
>
> *Q4. The paper notes occasional SDXL failures. Are there architectural insights explaining FLUX’s better compatibility?*
>
> [Response]
> This is an insightful question, and the architectural differences between FLUX and SDXL likely contribute to FLUX's better compatibility with our method.
>
> **FLUX's Architectural Advantages:** FLUX is a transformer-based model that utilizes a unified self-attention mechanism with modality-specific projection layers. This design allows FLUX to fuse semantic information across visual and textual modalities without relying on explicit cross-attention, leveraging the transformer's strength in modeling complex dependencies. In our method, this means that the newly injected subject visual tokens, text tokens, and current generated visual tokens are jointly considered within FLUX's unified attention layers. This simultaneous interaction likely enables FLUX to better balance subject consistency and prompt following, leading to improved performance.
>
> **SDXL's Architectural Considerations:** In contrast, SDXL is a U-Net-based model that combines convolutional and transformer blocks. Its transformer blocks contain separate cross-attention layers for text information and self-attention layers for visual information flow between patches. When we inject subject visual tokens into SDXL's self-attention layers, the interaction between these subject tokens and the text tokens (handled by separate cross-attention layers) might be less integrated. This separation could make it more challenging for SDXL to optimally balance text alignment and subject consistency, as the model needs to reconcile information from distinct interaction pathways.
>
> **General Model Scale and Performance:** Beyond architectural design, FLUX is generally a larger and more capable model for text-to-image generation compared to SDXL, which has been demonstrated in various experiments. The transformer architecture, in general, has proven more effective when scaling model size and dataset, contributing to FLUX's overall superior performance in text-to-image generation.
>
> *L1. Expand on potential misuse and mitigation strategies.*
>
> [Response]
> We acknowledge the importance of discussing potential misuse, especially for generative models. We briefly touch upon the positive societal impacts in Appendix A. We will expand on the "Broader Impacts" section (Appendix A) to include:
>
> Potential Misuse: Explicitly mention the risk of generating misleading or harmful content, such as deepfake stories or perpetuating stereotypes, given the enhanced consistency capabilities.
> Mitigation Strategies: While our work focuses on foundational research, we can suggest general mitigation strategies applicable to text-to-image models.  These include: Ethical Guidelines for Use, Watermarking/Provenance,Responsible Deployment
>
> *L2. Could the method handle longer story sequences.*
>
> [Response].
> We propose using a sliding window technique for the prompt lists. This approach allows the model to maintain context and subject consistency across a large number of images by processing prompts in overlapping segments. We conducted experiments using this solution, and the results demonstrate that our method can effectively generate consistent subjects throughout long stories and also maintaining strong text alignment.
> We will include these detailed results and a discussion of the sliding window technique in the revised version of our paper.

---

### Official Review · Reviewer_mtYN · 2025-07-05

**Clarity:** 2
**Significance:** 3
**Originality:** 2
**Rating:** 4
**Confidence:** 3

**Summary:**

This paper tackles the challenge of training-free consistent text-to-image generation. The authors propose a new sampling strategy called Asymmetry Zigzag Sampling (AZS). This method modifies the standard diffusion denoising process by decomposing each step into three sub-steps: zig, zag, and generation. This asymmetric design aims to disentangle the tasks of maintaining subject identity and adhering to the narrative prompt, thereby achieving a better balance between the two. The authors validate their method on both unet and dit based diffusion models.

**Questions:**

1. My questions on the impact, design choices, writing and ablation experiments are listed in the weakness session.
2. Some may claim this method introduces more computational overhead. I think it is acceptable as long as the strategy is applicable in general subject consistent generations. I am willing to raise my score as long as my concerns are cleared by the authors.

**Ethical Concerns:**

["NO or VERY MINOR ethics concerns only"]

**Final Justification:**

All of my concerns are well addressed. I would like to raise my score to 4: Borderline accept considering the overall contribution is limited and current writing quality.

**Limitations:**

yes

**Quality:**

3

**Strengths And Weaknesses:**

Strengths:
1. [Better Performance] The method is benchmarked against a wide range of relevant and strong baselines, including both training-free and training-based techniques. The result surpasses its close baseline 1Prompt1Story in all three metrics.
2. [Straight Forward Idea] Using test time scaling to strike the balance between text alignment and subject consistency is straight forward and effective

Weakness:
1. The idea of incorporating inversion process during sampling cannot be considered new, for example as in [1]. Actually, it can be considered as a special case of test time scaling [2], with a different guidance.
2. [Incremental based on 1Prompt1Story] The main framework heavily relies on 1Prompt1Story, which lessens the impact of the paper itself. Can this sampling strategy be applied to other consistency generation frameworks?
3. [Method clarity] The method part is hard to follow even after reading ConsiStory and 1Prompt1Story. I am not sure which designs of the previous work are adopted and which are not. I suggest the author to add details for the token caching, and unify the terms such as "one prompt", "SSC" etc., which are not described in the method section. More precise, step-by-step details would improve reproducibility.
4. [Lacks ablation studies] The ablation is designed to show each sampling process is necessary, but is not enough to answer all design choice questions.
    First I am curious how are the time step set in zig-gen and zig-zag, since they are 2-forward forward-backward, not aligned with conventional time steps.
    Second, and alternative zag process with SSC or one prompt should be given, to prove zag could fight identity overfitting
    Third, why adding gen step at last also improves clip-i?

[1] Zigzag Diffusion Sampling: Diffusion Models Can Self-Improve via Self-Reflection
[2] Scaling Inference Time Compute for Diffusion Models

---

> ### Author Rebuttal · Authors · 2025-07-30
>
> Thanks for raising all your concerns, below are our responses:
>
> *W1. The idea of incorporating inversion process during sampling cannot be considered new, for example as in [1]. Actually, it can be considered as a special case of test time scaling [2], with a different guidance.*
>
> [Response].
> We appreciate you highlighting related works on zigzag sampling and test-time scaling. We agree that zigzag sampling, as a concept for decomposing denoising steps, has been explored (e.g., [1]). Our novelty lies not merely in incorporating an inversion process, but in the asymmetric design of our Zigzag Sampling with Asymmetric Prompts and Visual Sharing (AZS) and its specific application to the challenging problem of maintaining subject consistency across multiple images in visual storytelling while preserving text fidelity.
>
> While "test-time scaling" [2] refers to a broader category of techniques that adjust model behavior during inference, our AZS is a concrete sampling strategy that orchestrates the flow of specific semantic information (visual subject cues and textual prompts) in an asymmetric manner across the zig, zag, and generation sub-steps. Previous zigzag methods primarily focused on improving general generation quality or text following. In contrast, our key innovation is the strategic and asymmetric injection of subject-specific visual information only in the "zig" (exploration) step, using a null prompt in the "zag" (refinement) step to prevent overfitting, and leveraging the full textual prompt in the "generation" step for narrative coherence. This precise control over information flow within the zigzag framework is what distinguishes our method and allows us to achieve a superior balance between subject consistency and prompt fidelity, which is a fundamental requirement for visual storytelling that prior methods struggle with. We will elaborate on this distinction in the revised introduction and method sections.
>
> *W2. The main framework heavily relies on 1Prompt1Story, which lessens the impact of the paper itself. Can this sampling strategy be applied to other consistency generation frameworks?*
>
> [Response]
> We conducted an experiment using our asymmetric zigzag sampling technique without the one-prompt method.
> clip-I: 0.9101, clip-T: 0.8876, DreamSim: 0.2113
> These results still outperform both the Consistory and StoryDiffusion models, demonstrating the generalizability of our proposed sampling strategy. Moreover, when combined with the one-prompt technique, it achieves the best performance, highlighting the complementarity between the two methods.
>
>
> *W3. The method part is hard to follow even after reading ConsiStory and 1Prompt1Story. I am not sure which designs of the previous work are adopted and which are not. I suggest the author to add details for the token caching, and unify the terms such as "one prompt", "SSC" etc., which are not described in the method section. More precise, step-by-step details would improve reproducibility.*
>
> [Response]
> Thanks for raising this issue and providing the suggestions.
> **Delineation of Adopted vs. Novel Designs**: We will explicitly state which components are adapted from prior work (i.e., , the idea of using an identity-focused prompt to extract subject-relevant visual features from [27], and the prompt fusion/reweighting mechanism from [28] for text fidelity) and clearly highlight our novel contributions (Asymmetry Zigzag Sampling, including Zig Visual Sharing (ZVS) and Asymmetric Prompt Zigzag Inference (APZI)).
> **Unifying Terminology**: We will standardize and clearly define all terms.
> "One Prompt" refers to the concatenated prompt strategy introduced by 1Prompt1Story [28], which we integrate for maintaining narrative coherence across scenes.
> "SSC" (Subject Semantic Control) refers to our proposed module (illustrated in Figure 3(d)) that performs the asymmetric visual guidance within the zigzag sampling process. Step-by-Step Details: We will provide a more precise, step-by-step algorithmic description ( with pseudocode) for the Asymmetric Zigzag Sampling process, detailing how ZVS and APZI are integrated into the zig, zag, and generation sub-steps. This will significantly improve reproducibility.
> W4. The ablation is designed to show that each sampling process is necessary, but is not enough to answer all design choice questions. First I am curious how are the time step set in zig-gen and zig-zag, since they are 2-forward forward-backward, not aligned with conventional time steps. Second, and alternative zag process with SSC or one prompt should be given, to prove zag could fight identity overfitting Third, why adding gen step at last also improves clip-i?
>
> *W4.1 how are the time step set in zig-gen and zig-zag?*.
>
> [Response]
> We apologize for the confusion caused by the "zig-gen" and "zig-zag" terminology in Table 3. This refers to where our proposed visual semantic injection is applied within the zigzag sampling sub-steps, not a different type of zigzag sampling. The core zigzag sampling mechanism (forward denoising for zig/generation, inverse denoising for zag, as defined in line 175) remains consistent across all these ablation settings. The "time step t" in these equations refers to the standard diffusion timestep.
> **"Asymmetry" (Our Method):** Visual injection (ZVS) only in zig step. Text guidance (APZI with full prompt) in zig and generation steps. Null prompt in zag step.
> **"Zig-gen":** Visual injection (ZVS) in zig and generation steps. Text guidance (full prompt) in zig and generation steps. Null prompt in zag step.
> **"Zig-zag":** Visual injection (ZVS) in zig and zag steps. Text guidance (full prompt) in zig and generation steps. Null prompt in zag step.
> **"All" (Fully Symmetric):** Visual injection (ZVS) in zig, zag, and generation steps. Text guidance (full prompt) in zig and generation steps. Null prompt in zag step.
> We will clarify this distinction in the revised paper, making it explicit that these ablations examine the impact of where visual conditioning is applied within our established asymmetric zigzag framework, not variations of the zigzag process itself.
>
> *W4.2 alternative zag process with ssc or one prompt should be given ?*
>
> [Response]
> This is a valuable suggestion. Our hypothesis is that using a null prompt in the zag step allows the subject-specific visual information injected in the zig step to propagate and "settle" into the latent representation without being immediately overwritten or overly constrained by new textual descriptions, thereby preventing overfitting to prompt specifics and improving generalizable subject consistency.
> We have conducted the following experiment:
> **"Asymmetry + Zag-Text"**: Our "Asymmetry" method, but with the full textual prompt (+zag-text) (or even the "one prompt" / "SSC" guidance) also applied during the zag step, instead of a null prompt.
> Asymmetry + zag-text         clip-I :0.9098     clip-T :0.8841       DreamSim :0.2251.
> Asymmetry + zag-null          clip-I :0.923       clip-T :0.8946        DreamSim :0.1789.
> The results above show that adding text in zag step leads to significant decreases in subject consistent evaluation metrics including clip-Ii and DreamSim score.
> The ablation of zag with ssc is exactly the experiment named zig-zag  in table 3 ( see above explanation of methods in table 3).
>
> *W4.3 why adding gen step at last also improve clip-i ?*
>
> [Response]
> There might be a slight misunderstanding here. In our "Asymmetry" method, the "gen step" (generation step) always uses the complete textual prompt for guidance. The improvement in CLIP-I (subject consistency) is not solely because of "adding the gen step at last." Rather, it is due to the synergistic effect of the entire asymmetric zigzag sampling process:
> **Zig Step (Exploration + Initial Identity Grounding)**: Strong subject-specific visual cues are injected, firmly establishing the subject's identity early on.
> **Zag Step (Refinement + Propagation, Null Prompt)**: The latent space is refined without additional textual or visual interference. The null prompt here is critical; it allows the visually conditioned subject information from the zig step to propagate through the denoising process without being immediately pulled too strongly by scene-specific textual prompts, which could otherwise lead to "identity overfitting" or reduce scene diversity.
> **Generation Step (Final Alignment, Full Text Prompt)**: This step then focuses solely on aligning the image with the specific textual prompt, leveraging the consistent subject information that has been preserved and propagated from the zig and zag steps. Because the earlier zig and zag steps have already done the heavy lifting of grounding the subject consistently and preparing the latent space, the generation step can effectively refine the image according to the narrative without losing subject identity.
>
> Therefore, the improvement in CLIP-I is a result of this **carefully orchestrated asymmetric information flow.** The "generation step" is where the narrative is finally realized with the subject consistently maintained due to the preceding zig and zag steps. The comparison in Table 3 specifically highlights that applying visual conditioning during the "generation" step (as in "Zig-gen") leads to poorer text alignment, while our "Asymmetry" method avoids this by focusing the "gen" step on text alignment alone after subject conditioning in the "zig" step.

---

> > ### Comment · Reviewer_mtYN · 2025-08-04
> >
> > Thank the authors for their responses to my concerns/questions. All of my concerns are well addressed. I would like to raise my score to 4: Borderline accept considering the overall contribution is limited and current writing quality.

---

> > > ### Author Response · Authors · 2025-08-05
> > >
> > > Thank you for your thoughtful follow-up and for taking the time to review our responses. We’re glad to hear that your concerns have been fully addressed.
> > >
> > > Regarding the score update, we appreciate your willingness to raise the score to 4: Borderline Accept. However, we kindly note that the change does not yet appear to be reflected in the review system. When convenient, could you please ensure that the score has been updated in the system?
> > >
> > > Thank you again for your time and consideration.

---

### Note · Authors · 2025-08-12

We sincerely thank the Area Chair and all reviewers (mtYN, 4vAP, yE2y, U3LM) for their time and insightful feedback. The discussion period has been exceptionally valuable, and we are confident that we have thoroughly addressed all raised concerns.

Based on the reviews, we will make several improvements to the final manuscript which we detailed in the rebuttal:

Strengthening Clarity and Reproducibility: We will add to  the methodology section with a step-by-step description of the algorithm in pseudocode, unify all terminology, and provide explicit implementation details (e.g., k=0.2, layer-wise injection) as requested by the reviewers. We will also provide qualitative figures to more clearly demonstrate our method's advantages, addressing a key point from Reviewer U3LM.

Adding New Experimental Evidence: As prompted by Reviewers mtYN and 4vAP, we have already conducted new ablation studies. The results, which will be added to the paper, empirically validate our core design choice and strengthen our claim that using a null prompt in the "zag" step is critical for balancing subject consistency and text fidelity.

Providing Deeper Analysis: We will integrate quantitative metrics for computational overhead, a more detailed architectural analysis explaining the performance differences between FLUX and SDXL, and a more nuanced discussion of failure cases in the appendix.

We were encouraged by the constructive dialogue following our rebuttal. We are grateful that Reviewer mtYN confirmed their concerns were 'well addressed' and raised their score, and that other reviewers acknowledged our planned clarifications. We are confident that our revisions, which incorporate the valuable feedback from all four reviews, will resolve any remaining reservations about clarity and contribution.

---

### Decision · Program_Chairs · 2025-09-17

**Decision:**

Accept (poster)

**Comment:**

Paper presents the approach for consistent training-free multi-image generation. Reviewers unanimously give a paper a Borderline Accept. Initial set of comments focused on (1) lack of / incremental novelty of the approach (e.g., with respect to 1Prompt1Story [mtYN] and others [4vAP]); (2) lacking clarity in the approach description [mtYN, yE2y, U3LM]; (3) lacking ablations [mtYN, 4vAP], (3) lacking motivation for some design choices [4vAP, yE2y, U3LM], (5) lack of analysis of inference time [4vAP, U3LM], and (6) lack of convincing qualitative results that illustrate clear advantages over the baselines [U3LM].

Rebuttal has provided many needed clarifications and reviewers mostly found it compelling, resulting in the current consistent Borderline Accept scores.

AC has read the comments, rebuttal and discussion that followed carefully, as well as  the paper itself. Similar to the reviewers AC finds the approach interesting, albeit limited in novelty. AC also agrees with [U3LM]  that qualitative results are a bit hit-and-miss, where some do show clear advantages, while others (Figure 6 bottom) hardly show an improvement and can even be considered inferior to 1Prompt1Story baseline. However, the overarching idea of asymmetric zig-zag sampling is compelling and could potential have broader use. On these grounds AC is leaning towards Acceptance.